# Conducting non-pandemic research during the COVID-19 pandemic: A case study and documentary review

Micah J. June[1,2], Mary I. Otiti[2,3], Nicodemus O. Mbanda[2], Polland O. Miruye[2], Stephen J. Allen[3], Dónal O'Mathúna[4]*

1 Department of Public Health, Pharmacology and Toxicology, University of Nairobi, Nairobi, Kenya,
2 Centre for Global Health Research, Kenya Medical Research Institute (KEMRI), Kisumu, Kenya,
3 Department of Clinical Sciences, Liverpool School of Tropical Medicine, Liverpool, United Kingdom,
4 College of Nursing and Center for Bioethics, The Ohio State University, Columbus, Ohio, United States of America

* omathuna.6@osu.edu (DO)

## Abstract

The COVID-19 pandemic presented unprecedented challenges to the conduct of non-pandemic research. The pandemic disrupted regulatory processes, strained health systems, and necessitated ethical recalibrations. This paper is a case study retrospectively examining the experience of conducting one non-pandemic clinical study of infant nutritional supplements in western Kenya between October 2020 and January 2024. The original study documents were reviewed to identify lessons learned, particularly those relevant to ethical and regulatory aspects of conducting non-pandemic research during a pandemic. A retrospective documentary case analysis was conducted using the archived ethics and regulatory submissions, feedback received during those submissions, study progress reports, meeting minutes, and relevant peer-reviewed literature. A framework analysis approach guided thematic coding, supported by independent dual review to ensure reliability (Cohen's Kappa ≥0.6). Public health restrictions during the third pandemic wave, particularly at the main recruitment site, compounded delays. Adaptive measures included reduced physical contact, use of satellite recruitment centers, community engagement with Traditional Birth Attendants, and remote monitoring via digital platforms. Parallel submission to both the Ethics Review Committee and the Regulatory Authority notably expedited the overall turnaround time for approval of protocol amendments. Despite these challenges, the study achieved an impressive 91% follow-up completion rate, with 96% of scheduled home visits successfully carried out. This case study highlights how flexible, community-informed, and ethically responsive research practices can sustain clinical trials under adverse conditions. Strong stakeholder collaboration, open communication, and proactive risk mitigation were essential to maintaining study integrity and protecting participant welfare. These findings underscore the need

**Data availability statement:** In response to the journal requirements, we have made more of our data available as supplementary files. These include the original codes developed as part of the thematic analysis (Supplementary Table 1.), a detailed history of the ethics review process (Supplementary Table 2.) and a similar history of the process through regulatory approval (Supplementary Table 1.). We have also added to the manuscript more details about these approval processes and the data analysis in response to reviewers' specific requests. Much of our original data consists of direct correspondence between the PROSYNK team and the research ethics committee, Kenyan regulatory body and the study sponsor. In response to our request to make such documents publicly available, these bodies replied that these are confidential documents addressing sensitive information. These communications often include identifiable details of committee members and sensitive internal deliberations, which according to their established research governance standards are not meant for public release without formal institutional approval. Access to the confidential data can be requested from these authorities. For the KEMRI Scientific and Ethics Review Unit (SERU), contact seru@kemri.go.ke; for the Kenyan Pharmacy and Poisons Board (PPB), contact cta@ppb.go.ke; and for the Liverpool School of Tropical Medicine (LSTM) Research Ethics Committee Secretariat, contact lstmrec@lstmed.ac.uk.

**Funding:** Funding for The PROSYNK trial was provided by the Liverpool School of Tropical Medicine (https://www.lstmed.ac.uk/) through a grant from the Children's Investment Fund Foundation (CIFF; https://ciff.org/). This grant (R-1806-02780) funded the work of MJ, MO, NM, PM and SA on the PROSYNK trial. The retrospective documentary case analysis reported here was funded through a training grant from the Fogarty International Center of the U.S. National Institutes of Health (https://www.fic.nih.gov/). This grant (Award Number R25TW012217) funded the work of MJ and DO. The content of this article is solely the responsibility of the authors and does not necessarily represent the official views of the National Institutes of Health. The funders had no role in study design, data collection and analysis, decision to publish, or preparation of the manuscript.

**Competing interests:** The authors have declared that no competing interests exist.

for future research designs to embed contingency planning, community partnerships, and regulatory adaptability to ensure continuity and resilience for research conducted during public health emergencies.

## 1.0. Introduction

Supportive research environments are critical for enabling high-quality, impactful studies while maintaining ethical integrity and fostering professional growth. The onset of the COVID-19 pandemic posed significant challenges to the global research ecosystem, particularly for non-pandemic-related research [1]. Ethical research guided by principlism, including respect for autonomy, beneficence, non-maleficence, and justice [2], became more complex in a context marked by public health emergencies, shifting vulnerabilities, and constrained resources. In some settings COVID-19 presented a humanitarian situation requiring additional considerations for undertaking research that included examining the need to promote the health of communities, respect their dignity and uphold their rights [3]. Conducting non-pandemic research during a pandemic raises important questions about the prioritization of resources, public health responsibilities, and the distribution of benefits and burdens across society [4]. While pandemic research should be prioritized, other types of research remain important as they contribute to other important health needs. When non-pandemic research has already been initiated, stopping that research could waste resources already invested in the study, or leave participants at a disadvantage or risk of harm from receiving only part of the planned intervention. Careful balancing of the risks and benefits is needed before terminating, delaying or not initiating non-pandemic research during a pandemic.

The pandemic required a realignment of research priorities and careful trade-offs in decision-making. The World Health Organization (WHO) [5] emphasized the need to reassess ongoing and planned research projects, urging researchers and regulatory bodies to consider whether the risks posed by study participation in a pandemic context were ethically justifiable. Consequently, non-COVID-19 research was subjected to heightened scrutiny, new regulatory frameworks, and logistical challenges ranging from delays in ethical approval to disruptions in supply chains and field activities.

In Kenya, institutions such as the Kenya Medical Research Institute (KEMRI) and the Pharmacy and Poisons Board (PPB) implemented interim guidelines that prioritized COVID-19 studies, revised consent procedures, minimized physical interactions, and adapted protocol requirements [6]. These changes, while necessary, had unintended implications for non-pandemic clinical trials. Studies not directly addressing COVID-19 faced delays, resource reallocation, and field constraints that threatened their methodological rigor and ethical compliance [7].

The COVID-19 pandemic disrupted research systems worldwide, exposing vulnerabilities but also catalyzing opportunities for resilience, adaptability, and transformation. Drawing on Walker et al.'s framework of resilience theory, research processes

can be understood in terms of their ability to absorb shocks and maintain core functions (resilience), adjust practices within existing structures (adaptability), or reconfigure practices into new operational models (transformability) [8]. This resilience lens is particularly relevant to addressing ethical issues that arise with clinical trials during pandemics, as seen in particular with COVID-19 where investigators and institutions were compelled to sustain recruitment ethically, ensure ethical oversight, and innovate through remote technologies and decentralized designs while upholding ethical commitments. Positioning research responses within this resilience framework provides a structured way to examine not only how systems coped with disruption, but also which adaptations or transformations maintained ethical commitments and should be sustained in the post-pandemic landscape.

This paper presents a retrospective documentary case analysis of a non-COVID-19 clinical trial conducted during the pandemic in western Kenya: namely, the PRObiotics and SYNbiotics to improve gut health and growth in infants in western Kenya (PROSYNK trial). This randomized controlled trial, conducted from October 2020 to January 2024, investigated the efficacy of probiotics and synbiotics in improving gut health and reducing systemic inflammation in infants [9]. Despite its non-pandemic focus, the study was shaped profoundly by the pandemic context. The documentary analysis reported here aims to highlight the ethical challenges and trade-offs encountered, as well as the adaptive strategies employed to maintain research quality, participant safety, and ethical standards in a dynamic and constrained environment.

## 2.0. Materials and methods

The PROSYNK trial was conducted through a multidisciplinary collaboration between Kenyan and international institutions. Local leadership and day-to-day trial coordination were provided by investigators at the Kenya Medical Research Institute – Centre for Global Health Research (KEMRI-CGHR) in Kisumu (MO), ensuring contextual relevance and integration within the existing research infrastructure. Oversight and trial design support were contributed by the Liverpool School of Tropical Medicine (LSTM), United Kingdom (UK), including extensive expertise in infectious disease research and clinical trial methodology (SA). Microbiome expertise was provided by the Quadram Institute Bioscience, UK, whose team advised on the scientific rationale for probiotic and synbiotic interventions and laboratory analyses.

This collaborative structure combined local implementation capacity (MJ, PO, NM) with international methodological and laboratory expertise, allowing the trial to be scientifically rigorous, contextually grounded and ethically rigorous. The integration of KEMRI-CGHR's established community networks, LSTM's trial governance systems, and the Quadram Institute's specialist laboratory facilities underpinned the feasibility and scientific integrity of the trial [9].

### 2.1. Ethics statement

The original PROSYNK trial received ethics approval from the KEMRI Scientific and Ethics Review Unit (SERU) on 16 March 2020 (KEMRI/SERU/CGHR/320/3917) and the Kenyan PPB on 5 August 2020 (ECCT/20/04/02). LSTM agreed to act as study Sponsor on 7 August 2020 (ref 19-048). To conduct the retrospective analysis reported here, an amendment was submitted to SERU and approved on 24 March 2025. A Data Sharing Agreement to access archived PROSYNK records was signed between the first author and LSTM on 26 January 2025. Only anonymized archived data were accessed on 31 March 2025. No participant-identifiable information was accessed, and all data handling adhered to established ethical and confidentiality standards.

#### 2.1.1. Ethics and Regulatory Approval during the COVID-19 pandemic. In Kenya, clinical trials require approval from a nationally accredited ethics committee followed by authorization from the PPB, which under normal conditions aims to respond within 30 working days. During the COVID-19 pandemic, the PPB introduced emergency guidelines enabling expedited and parallel ethics and regulatory review, remote submissions, and flexible protocol amendments to protect participants [6]. These measures reflected WHO guidance that research in emergencies must remain scientifically valid and ethically sound [5].

## 2.2. Study setting

The methods and results of the PROSYNK trial have been published previously [9]. Only those aspects of the clinical trial relevant to the documentary analysis are reported here. Six hundred newborns under four days old were enrolled from Homa Bay County Teaching and Referral Hospital (HBCTRH) and nearby delivery centres in western Kenya. The infants were randomly assigned, with stratification based on HIV exposure, to one of four study groups in equal proportions (1:1:1:1) to receive either one of two synbiotics, a probiotic, or no supplement. Participants were scheduled to receive 32 doses of the assigned intervention over six months, administered through directly observed therapy during home visits. The assigned supplements were administered daily for the first 10 days, then weekly up to six months of age. The participants were monitored until they reached two years of age. The primary outcome measure was systemic inflammation at six months, evaluated using plasma alpha-1-acid glycoprotein levels.

HBCTRH is located in Homa Bay town, the administrative and commercial centre of Homa Bay County in western Kenya. The hospital serves as a key referral centre for maternal and child health services in a region with high infant morbidity and limited healthcare access. The geographical setting, characterized by rural catchments and socio-economic vulnerabilities, was critical in shaping the study's design and response during the COVID-19 pandemic.

## 2.3. Study design

The study reported here employed a retrospective documentary case analysis and reflective reporting methodology to explore the operational, ethical, and logistical experiences of conducting a non-COVID-19 clinical trial during a global pandemic. The approach enabled in-depth examination of the adaptations made in the midst of evolving regulatory, health system, and community-level challenges.

## 2.4. Case selection

The PROSYNK trial was purposefully selected due to the first author's direct involvement in its implementation. This involvement spanned regulatory submissions, community engagement, participant recruitment, follow-up procedures, and dissemination of findings. The PROSYNK trial was conducted amid heightened pandemic constraints and thus provided a suitable case for examining resilience and innovation in research conduct under crisis conditions.

This single case study was chosen to capture detailed insights into conducting a clinical trial during the COVID-19 pandemic. This design allows exploration of ethical, regulatory, and operational processes in context, offering depth that broader approaches might not capture. While findings are context-specific, they provide analytical generalization, highlighting strategies such as adaptive regulation, community engagement, and infection prevention that can inform trials in other low-resource or emergency settings.

## 2.5. Data collection and analysis

Archived PROSYNK trial documents were retrieved from a central archival facility at KEMRI-CGHR in Kisumu. Key materials retrieved and reviewed included ethics and regulatory submissions, feedback provided during those submissions, study Gantt charts, meeting minutes, progress reports, and sponsor communications. A chronological timeline of events was constructed and compared against the original project plan to map deviations and identify delays.

Documents were included if they directly informed the planning, approval, or implementation of the clinical trial during the COVID-19 pandemic. These comprised ethics and regulatory submissions, meeting minutes with stakeholders, institutional and national COVID-19 guidelines, standard operating procedures (SOPs), and correspondence with oversight bodies. Documents were excluded if they were unrelated to trial conduct, duplicates of already captured materials, or lacked sufficient detail to inform analysis. This approach ensured that only sources relevant to understanding the ethical, regulatory, and operational dimensions of trial implementation during the pandemic were retained.

Additional data on the broader pandemic context such as health worker strikes and containment measures were gathered from peer-reviewed publications and mainstream news sources. All documents relevant to the research questions were logged systematically using a Document Access Log. Triangulation was achieved by comparing data across ethics and regulatory submissions, SOPs, stakeholder meeting minutes, and internal reports, alongside external sources such as news reports and published articles. This integration of internal and external perspectives reduced reliance on any single source and provided a more robust understanding of ethical, regulatory, operational, and societal influences on trial implementation during COVID-19.

Thematic analysis followed a structured framework adapted to the clinical trial implementation context. Documents were read and re-read independently by three researchers, with original codes noted in Excel spreadsheets (see S1 Table). No specific data analysis software was used. From the original codes, five overarching themes were developed and guided further coding: (i) regulatory and ethics oversight, (ii) healthcare system changes, (iii) adaptive recruitment and retention strategies, (iv) public health and socio-economic adjustments, and (v) overall trial performance (Table 1).

Data were manually coded in iterative cycles. Initial open coding generated descriptive categories, which were subsequently refined into five focused themes. Further codes were inductively added as new patterns emerged and included within the five themes (e.g., communication with research ethics committee (REC), community trust/mistrust, ethical tensions; see S1 Table).

**Table 1. Framework for thematic analysis of the PROSYNK trial case study.**

| Theme | Subtheme | Summary Findings |
|---|---|---|
| Regulatory and Ethics Oversight | 1. Approval turnaround during pandemic 2. COVID-19 amendment requirements 3. Adaptive strategies for compliance | Initial ethics submission: 17 Jun 2019. COVID-19-related protocol amendment resubmitted 15 Jan 2020; approval 16 Mar 2020. Regulatory submission: 20 Apr 2020; approval 5 Aug 2020. Adaptive strategies included timely responses to queries, protocol revisions, and alignment with COVID-19 guidance. |
| Healthcare System Changes | 1. Healthcare worker (HCW) strikes 2. Facility guidance on patient visits | Two HCW strikes occurred during the pandemic. Declines in service utilization observed. Homa Bay facility issued intermittent closures due to COVID-19 staff exposure. Study adapted with staff work cohorts and routine testing. |
| Adaptive Recruitment and Retention Strategies | 1. Recruitment innovations 2. Retention measures | Recruitment expanded to satellite clinics; collaboration with Traditional Birth Attendants (TBAs) supported referrals. Remote follow-up was incorporated via protocol amendment. Staff organized in cohorts; Personal Protective Equipment (PPE) provision, routine testing, and CommCare visit-tracking supported trial continuity. |
| Public Health & Socio-economic Adjustments | 1. Preparedness and response 2. Government restrictions | Kenya's pre-pandemic preparedness was moderate. Pandemic response improved with vaccination, oxygen access, and public-private partnerships. Lockdowns and curfews disrupted recruitment and follow-up. Despite this, 600 participants were enrolled (28 Oct 2020–13 Jan 2022), with 91% completing follow-up. Scheduled visit and sampling completion exceeded 90%. Unscheduled visits: 7,816. Severe adverse events: 102, all reported per guidelines. |
| Overall Trial Performance | 1. Retention and outcomes 2. Quality and oversight | The trial maintained integrity despite the pandemic: 18,412 successful home visits for supplement administration; 93% of scheduled clinic visits completed; 93% primary outcome documented; no critical audit findings. Key enablers included protocol flexibility, community collaboration, proactive sponsor oversight, and staff commitment. |

## 2.6. Reliability and validity

To ensure analytical rigor and consistency in data interpretation, systematic document review was employed. Two trained research assistants independently examined all documents using a pre-defined, standardized coding framework designed to capture relevant themes, patterns, and categories systematically. This dual-review approach aimed to minimize individual bias and enhance the reliability of the coding process.

To assess the level of agreement between the two reviewers, Cohen's Kappa statistic was calculated. This statistical measure accounts for agreement occurring by chance and provides a more robust evaluation of inter-rater reliability. A Kappa value greater than 0.6 was considered indicative of an acceptable level of agreement, reflecting substantial consistency between coders.

In instances where discrepancies arose between the reviewers, the documents in question were flagged for further analysis. The principal investigator then conducted an independent review of the contested items. Following this, the research team held structured debriefing discussions to resolve differences. Through these discussions, consensus was reached, ensuring that the final coding accurately reflected a shared interpretation of the data. This iterative process contributed to the methodological rigor and reliability of the study's qualitative analysis.

## 3.0. Results

### 3.1. Regulatory timelines and study implementation

The initial ethics submission for the PROSYNK trial was made to SERU on 17 June 2019. The initial reviewers' response on 16 October 2019 required a major modification to one of the nutritional supplements to avoid potential exposure to cow milk protein by those taking the supplements. Addressing this formulation issue delayed ethics resubmission until 15 January 2020 and hence during COVID-19. The implementation timeline was therefore calculated from this second submission. Ethics approval for this submission was received on 16 March 2020. However, further amendments to the protocol and ethics application were required due to new COVID-19 regulations. The issues to be addressed ranged from clarification of recruitment strategies, participant reimbursement, and approval for future use of stored samples (further details in S2 Table). Regulatory submission to PPB was made on 20 April 2020, with approval received on 5 August 2020 (further details in S3 Table). Parallel submissions were permitted so that responses to the ethics committee's and regulatory authority's comments were made simultaneously during this period. Final SERU ethics approval was received on 14 July 2020. Following both approvals, recruitment began on 28 October 2020 after staff training on the approved protocol. Both ethics and regulatory approvals each took an average of six months; this was consistent with expectations.

In line with Ministry of Health (MoH) guidelines on COVID-19 infection prevention and control (IPC), the study implemented several safety measures. Staff received IPC training, were provided with appropriate PPE, and adhered to physical distancing requirements consistent with MoH and WHO recommendations. Both home and clinic visits were conducted by trained and vaccinated personnel, and structured reporting procedures were established. These measures were reviewed as an amendment and approved by the ethics committee on 10 November 2020.

The trial was designed to complete recruitment within one year and follow participants for two years. The initial completion date was expected to be June 2023. With initial participant recruitment delayed until October 2020, the revised date for final data collection was October 2023. Further delays led to data collection continuing until 18 January 2024, representing a seven-month delay compared to the original schedule (Fig 1).

The most substantial disruption occurred during Kenya's third wave of COVID-19 (April–July 2021). During this period, the study sponsor, LSTM, issued enhanced safety guidelines via a memo dated 21 June 2021. These measures were consistent with the Kenyan MoH COVID-19 protocols. As documented in study meeting minutes, prior to initiating the PROSYNK trial, key stakeholders engaged in transparent, culturally sensitive consultations, including community meetings, educational sessions, and feedback mechanisms that built trust in the trial ecosystem [10]. A Community Advisory Board (CAB) provided ongoing engagement between the community and research team, meeting quarterly to address

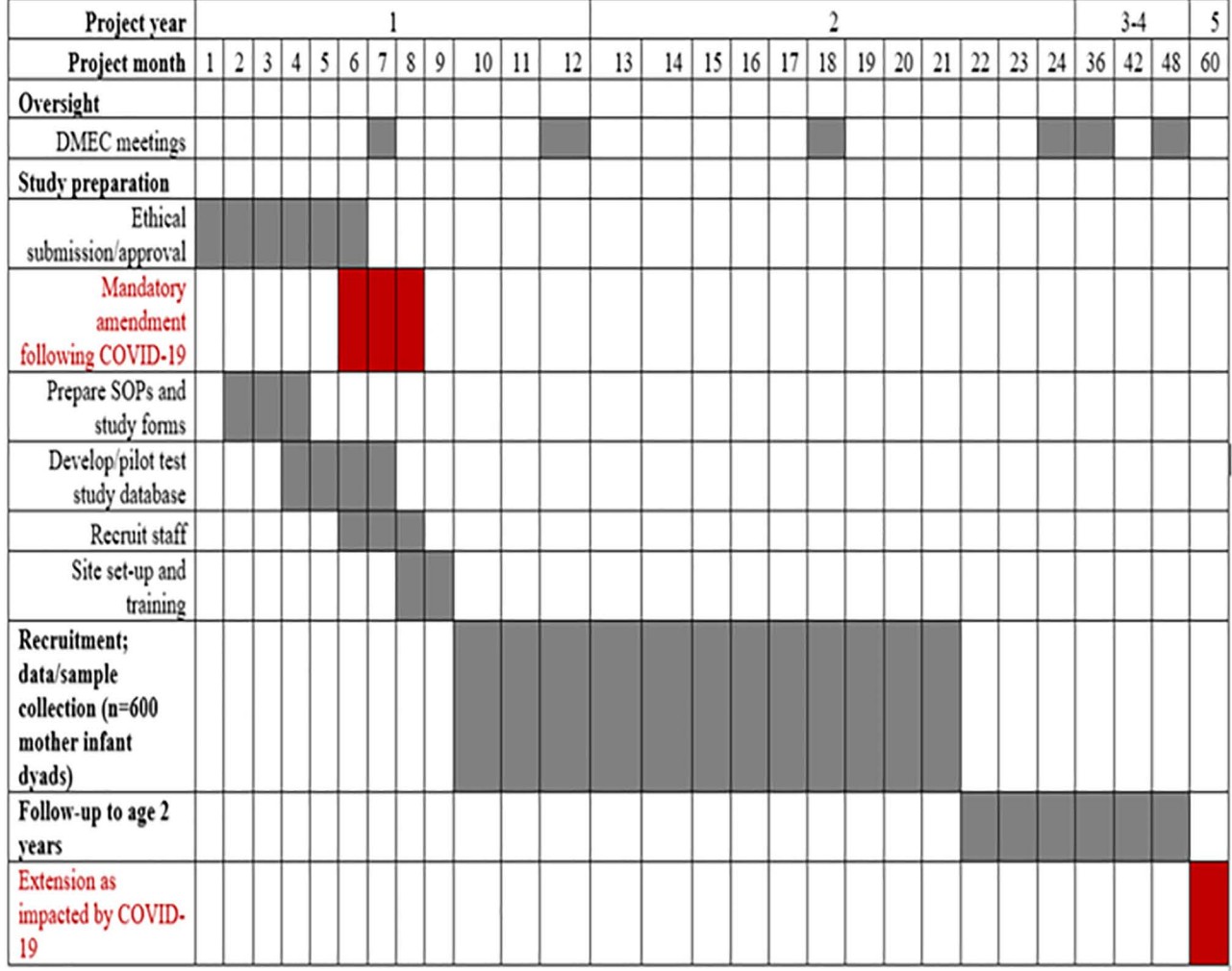

**Fig 1. PROSYNK Study Gantt Chart as impacted by COVID-19.**

concerns and advise on potential threats to trial conduct. Regular discussions with community representatives helped alleviate fears and communicate protective measures for participants and their households, thereby improving participation and recruitment in a community relatively unfamiliar with clinical research.

### 3.2. Recruitment and participant follow-up

HBCTRH, the primary recruitment site for the trial, adopted more stringent COVID-19 containment protocols than surrounding facilities. Operational challenges intensified during the third wave of COVID-19 in Kenya (April–July 2021), with mobility restrictions, infection control mandates, and periodic closures of hospital departments following staff exposure. HBCTRH temporarily closed due to confirmed COVID-19 cases among staff. With a shift in deliveries to peripheral sites and an increase in births attended by TBAs, participant recruitment was adversely impacted (Fig 2). Weekly reports submitted to the study sponsor reflected a consistent shortfall in recruitment targets.

Despite recruitment challenges, the study achieved a high follow-up completion rate: 546 out of 600 participants (91%) completed the study (Fig 3). Overall, 19,200 home visits were planned, but those for the control group were substituted

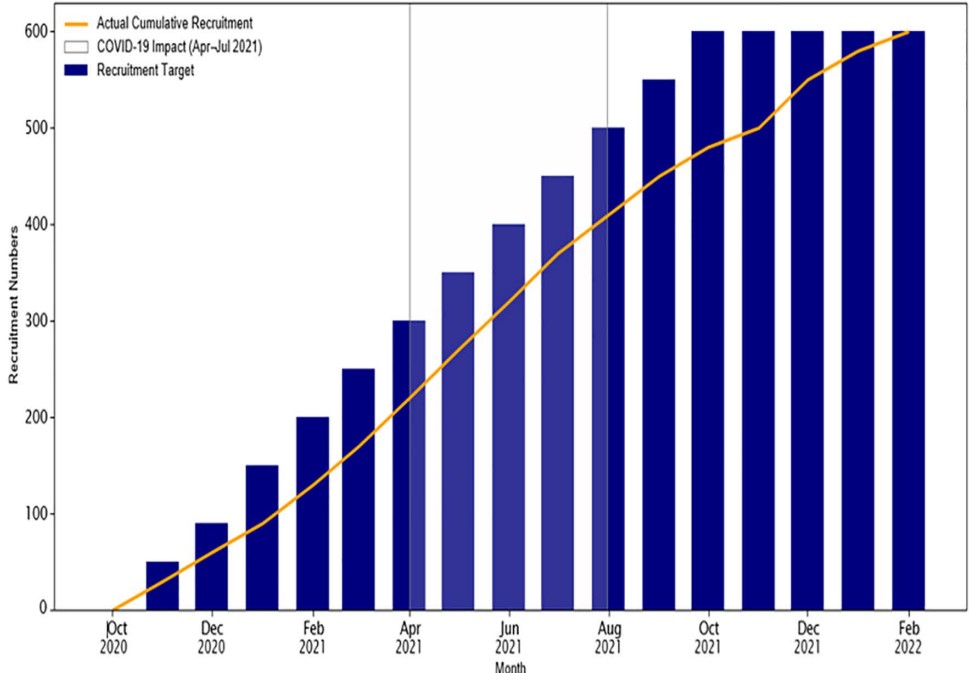

**Fig 2. Monthly Recruitment vs. Target as impacted by COVID-19 containment measures.**

with phone calls. Despite this adjustment, a total of 18,412 visits/calls, representing 96% of the planned interactions, were successfully completed. Overall, the proportion of completed visits/calls during the COVID-19 third wave was similar to that for the non-COVID-19 period. These visits, whether in-person or by phone, were used to administer supplements and/or deliver health messaging across both the intervention and control groups.

### 3.3. Healthcare and socioeconomic impacts of the pandemic

COVID-19 containment policies significantly restricted access to healthcare facilities, complicating study recruitment and follow-up. Peer-reviewed studies and media reports documented nationwide health system strain, high infection risk among healthcare workers, and resulting industrial action [11]. These factors, combined with localized COVID-19 outbreaks, led to intermittent departmental closures at the main recruitment centre, as noted in a news article dated 8 June 2021 [12]. Such disruptions underscored the volatile research environment during the pandemic period.

### 3.4. Adaptive strategies employed by the study team

To mitigate pandemic-related disruptions and ensure continuity of study, the PROSYNK team implemented a series of adaptive measures, documented in ethics and regulatory submissions, meeting minutes, and study reports. These strategies aimed to protect participants and staff while maintaining methodological and ethical rigor. The initial study protocol gave room for the addition of satellite facilities to help drive recruitment through deployment of outreach teams. Introduction letters and flyers were circulated to the peripheral facilities to facilitate referral of mothers due for delivery to the trial facility.

The minutes of frequent study leadership meetings documented the evaluation of prevailing COVID-19 conditions over time. Recommendations were shared with the Data Safety and Monitoring Board. The board advised on the course of the study following careful evaluation of risks to participants and staff.

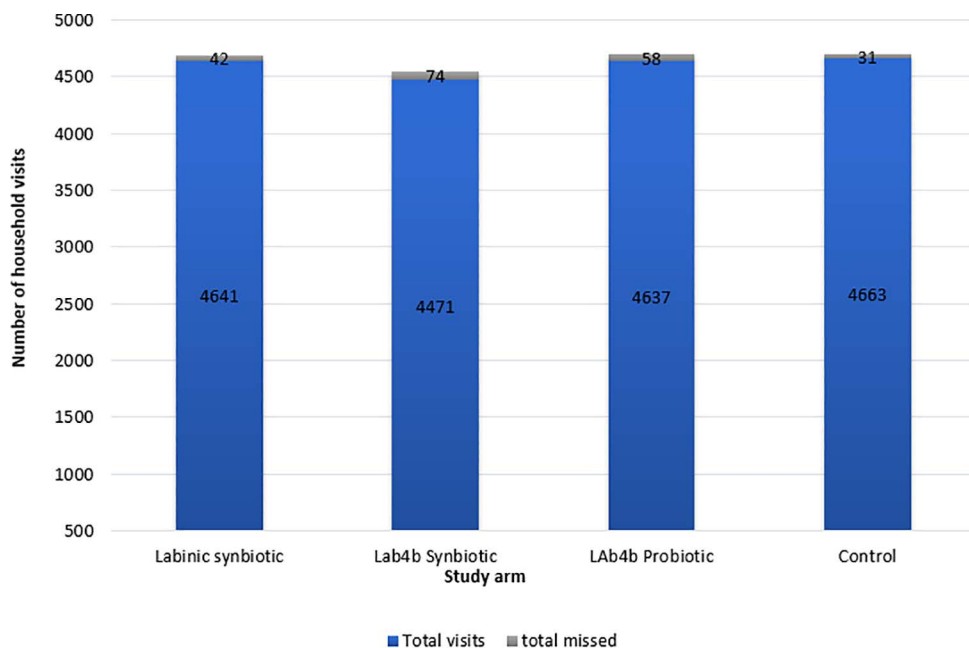

**Fig 3. Supplement administration and missed visits by study arm.**

Consultative meetings were held with community focal persons and enlisted TBAs, resulting in an agreement to support appropriate referrals of expectant mothers to the trial facility. The study team also engaged an active CAB to facilitate ongoing community dialogue. Minutes from these meetings highlighted the team's efforts to dispel myths surrounding COVID-19 and to communicate additional safety measures implemented to protect both participants and the wider community. For example, on 31 July 2020 a mandatory amendment was submitted to SERU in accordance with updated COVID-19 guidelines for research studies. The amendment was for the use of phone calls, in place of home visits, for participants in the control arm where supervised dosing was not required.

Clear communication was maintained through internal memos outlining the specific measures to be implemented due to COVID-19. To adapt to the evolving situation, the study team adjusted the enrollment target to a more manageable number, reducing it from twelve to six participants per week. Although this slowed recruitment, it was viewed as important given the pandemic context (Fig 2). Additionally, research staff operated in rotating teams, a strategy that helped minimize crowding at the research center. The work teams were balanced with each comprised of study team members able to conduct all study procedures adequately. The study coordinator ensured smooth transitions between outgoing and incoming teams by prioritizing all pending work assignments.

In addition to prioritizing the standard of care for all participants, the study team facilitated COVID-19 testing for individuals presenting with symptoms of severe acute respiratory illness. As the host institution's laboratory was an approved COVID-19 testing center, the study team was able to ensure a rapid turnaround of results to support timely clinical decision-making and patient care. Additionally, PPE was provided to general hospital staff and remedial measures were undertaken to curb the spread of COVID-19, including routine COVID-19 testing for study staff during work periods and disinfection of workspaces.

Study tasks and activities were carefully sequenced, with timelines for procedures such as visiting windows considered to ensure smooth implementation. Communication channels were established to support effective collaboration and timely

sharing of information among team members. The CommCare App was utilized to monitor scheduled clinic visits and track when supplement doses were due for participants.

## 4.0. Discussion

This case study of the PROSYNK trial provides insight into the realities of conducting clinical research in a low-resource setting during the COVID-19 pandemic. The trial faced many challenges, operational disruptions, and the need for rapid protocol adaptation, while relying on context-specific strategies such as culturally sensitive community engagement, flexible operational planning, and evolving infection prevention measures, to sustain continuity and ethical rigor. Situating these findings within the broader literature on research ethics and resilience during public health emergencies underscores both shared vulnerabilities and locally driven solutions. Our findings resonate with the global experiences documented by others who found that RECs adapted procedures during COVID-19 but often struggled to balance urgency with rigor [13]. In the PROSYNK trial, the Kenyan regulatory environment responded with revised guidelines from the PPB, but the timeline from resubmission in January 2020 to ethics approval in March 2020 and PPB approval in August 2020 reflects the cautious balance between speed and procedural integrity.

These shifts required by the pandemic demanded not only technical and logistical flexibility but also careful balancing of core ethical principles. Safeguarding autonomy through ongoing consent discussions, upholding beneficence and non-maleficence while modifying trial procedures, and ensuring justice in access to participation and care required deliberate negotiation alongside system adjustments. Positioning research responses within this dual lens of resilience and principlism provided a structured way to evaluate how trials can withstand disruption, how ethical commitments were preserved under pressure, and which adaptations or transformations should be sustained beyond the pandemic.

### 4.1. Operational disruption

The PROSYNK trial experienced significant delays that extended its closure date from the originally planned June 2023 to January 2024. These delays disrupted the overall study timeline and reflected broader global patterns of regulatory slowdown during the COVID-19 pandemic (Fig 1). Ethics review boards and regulatory authorities were widely reported to have reallocated attention toward COVID-19-related research priorities [5,7,13]. For the PROSYNK trail, operational disruptions were most acute during Kenya's third wave of COVID-19 (April–July 2021), when government-imposed mobility restrictions, infection control protocols, and intermittent hospital closures directly impacted research activities. For example, on 8 June 2021, entire departments at HBCTRH, the study's primary recruitment site, were temporarily shut down following confirmed staff infections [12]. These events echoed national trends in healthcare system strain, including labour unrest and a resurgence of traditional birth practices, as documented in Coastal Kenya [14].

Further, healthcare workers' strikes in Kenya during the COVID-19 pandemic significantly compounded the public health crisis. Most medical personnel in public hospitals engaged in industrial action, disrupting essential health services at a critical time. The strikes were driven by demands for adequate insurance coverage and sufficient PPE, both of which were lacking. These protests occurred amid rising COVID-19 cases and mounting pressure on the health system. The situation was further exacerbated by the deaths of at least 14 doctors from COVID-19, underscoring the urgent need for improved occupational safety and support for frontline health workers [15].

Within this context, achieving recruitment targets became increasingly unfeasible. These challenges highlight the fragility of facility-based studies during public health emergencies and underscore the need for adaptive research designs that can withstand future operational disruptions.

### 4.2. Adaptive and ethical research management

Despite the considerable operational challenges faced by the PROSYNK trial, the research team exhibited exemplary ethical leadership through resilience and adaptability. Key mitigation strategies were implemented to uphold the study's

integrity and participant welfare, including expanding recruitment efforts to satellite health centres, collaborating closely with community focal persons and TBAs, rotating staff teams to reduce COVID-19 exposure risk, and strengthening infection prevention protocols. These measures reflect core ethical principles of beneficence, justice, and respect for persons [16] ensuring that participant safety and equitable access remained paramount even amid a public health crisis. Shifting some visits, particularly for the control arm, to remote monitoring while maintaining in-person engagement for critical health assessments was guided by equity considerations, ensuring that participants in the control group continued to receive appropriate support and information, and that no group was disproportionately disadvantaged by the modified procedures.

Central to this ethical leadership was the deliberate focus on community trust-building. The team maintained frequent, transparent communication with healthcare workers, community stakeholders, and oversight bodies such as the study sponsor. This openness helped counter misinformation, foster accountability, and support informed, ethically sound decision-making throughout the study. As Mwangi et al. highlight, such strong leadership combined with inclusive dialogue is vital for sustaining research momentum and ethical rigor in times of uncertainty [17]. This approach underscores how ethical leadership can effectively navigate complex challenges, balancing scientific goals with the moral imperative to protect and respect participants and communities during crises.

### 4.3. High follow-up and participant retention

The study notably achieved a 91% follow-up completion rate and a 96% success rate in scheduled home visits/calls, demonstrating the commitment and resilience of the field teams. A key factor contributing to these outcomes was the integration of remote follow-up options for the control group, alongside the use of digital platforms such as CommCare. These technological adaptations facilitated uninterrupted data collection while reducing health risks associated with in-person contact. Such innovations highlight the critical role of flexible research designs and digital tools in maintaining data integrity and study continuity amid operational constraints, aligning with broader evidence on the benefits of digital health technologies in research during public health emergencies [13,18,19].

Priority home visits were conducted for participants who had been scheduled for phone follow-up but could not be reached. Consistent with findings by Houlding et al., limited access to mobile phones was observed among participants. These challenges highlight the need to develop a structured framework for remote monitoring in research, particularly in low-resource settings [20].

The implementation of COVID-19 control measures in Kenya varied notably across regions, shaped by local epidemiological trends, socio-economic conditions, and administrative frameworks. In rural areas such as the setting for this study, the enforcement of these measures was relatively less stringent, which allowed for the continued progression of research activities with minimal disruption [21].

### 4.4. Ethical responsiveness in public health emergencies

The study team demonstrated a proactive and ethically grounded response to emerging public health threats, going beyond protocol modifications. The provision of COVID-19 testing for symptomatic participants and staff, as well as routine testing, provision of PPE, and regular disinfection of workspaces, underscore a strong commitment to the safety and wellbeing of both research personnel and community participants, an often underreported but critical aspect of conducting research during public health crises. As Moodley et al. suggest, leadership in healthcare and research must proactively address systemic barriers to both service delivery and study participation during emergencies [22].

### 4.5. Strengths and limitations

This study provides a rare, in-depth account of clinical trial implementation in a low-resource setting during the COVID-19 pandemic. The use of anonymized trial documents, supplemented with meeting minutes, memos, and correspondence,

enabled detailed reconstruction of ethical, regulatory, and operational decision-making. Triangulation with published literature and media reports further enhanced interpretive validity and provided broader context beyond the trial itself.

The analysis has limitations as it relied primarily on documentary evidence, which may not capture all perspectives, particularly informal or undocumented challenges. Only records subjected to routine monitoring and archiving were included, and these were assumed to be accurate, complete, and contemporaneous. The focus on a single trial limits generalizability; findings are specific to the PROSYNK trial. While the insights generated may offer analytical generalization, they should be interpreted with caution when applied to other settings.

The authors acknowledge that their interpretation of the documents was shaped by their roles within and proximity to the PROSYNK trial. To mitigate potential bias, themes were derived systematically using a predefined framework and validated across multiple data sources. Nonetheless, the researchers' positionality may have influenced their emphasis on particular challenges or innovations, and this should be considered when interpreting the findings.

## 5.0. Conclusion and recommendations

The PROSYNK trial illustrates both the vulnerability and resilience of clinical research in the face of global health emergencies. While delays caused by public health restrictions disrupted the original study timeline, the research team's adaptive strategies grounded in ethical responsiveness, community engagement, and operational flexibility enabled the study to meet its core objectives. Recruitment that met original targets and high participant retention rates with follow-up to study completion, despite significant constraints, affirm the effectiveness of decentralized recruitment models, strong stakeholder collaboration, and technological innovations.

These findings highlight critical lessons for sustaining research in dynamic, resource-limited, or crisis-affected settings. Researchers, ethics bodies, and sponsors alike must be prepared to adjust traditional processes to uphold ethical standards while maintaining scientific rigor. The PROSYNK experience underscores the value of proactive, community-centric, and flexible research design, particularly when navigating the uncertainties of public health emergencies.

To strengthen future research implementation during pandemics or similar disruptions, the following recommendations are proposed:

1. Regulatory Flexibility with Ethical Vigilance

Ethics review committees should consider allowing the continuation of non-pandemic research where clear benefits exist and risks to participants, staff, and the broader community are demonstrably mitigated. A nuanced, risk-benefit approach is essential to avoid blanket suspensions that may inadvertently harm vulnerable populations.

2. Transparent Stakeholder Engagement

Researchers should foster regular, open communication with all stakeholders including participants, community leaders, sponsors, and oversight bodies to promote trust, reduce misinformation, and support ethical decision-making under evolving conditions.

3. Proactive Risk Monitoring and Response

Research teams must remain responsive to emerging threats by monitoring local trends and implementing context-appropriate mitigation strategies. Ensuring participant and staff safety through timely adjustments—such as remote follow-ups or PPE provision—can preserve study integrity while minimizing harm.

4. Flexible and Decentralized Research Models

Incorporating contingency plans into study protocols enhances resilience. The successful use of satellite sites, community health workers, and TBAs in PROSYNK demonstrates the value of decentralized recruitment strategies. Embedding such models at the design stage can improve reach and retention, especially in underserved or disrupted areas.

These recommendations advocate for a more agile approach to research in complex environments that remains ethically grounded. As the global research community continues to contend with uncertainty and rapid change, the lessons from PROSYNK serve as a roadmap for future preparedness, continuity, and ethical excellence in clinical research. This case study illustrates both the vulnerabilities and adaptive strengths of conducting research during a global health emergency. Despite operational disruptions, ethical responsiveness, community engagement, and operational flexibility enabled the study to achieve its objectives. These lessons, though grounded in a single case study, carry broader relevance for research in other low-resource, crisis-affected, or dynamic settings. They reinforce WHO [5], Council for International Organizations of Medical Sciences (CIOMS) [23] and other guidance on the importance of regulatory flexibility coupled with ethical vigilance, transparent stakeholder engagement, proactive contingency planning, and rigorous infection prevention and monitoring. Embedding decentralized and community-centric models into trial design, while strengthening oversight capacity, can help ensure continuity, equity, and ethical integrity of research in future public health emergencies.

## Supporting information

**S1 Table. Codebook mapping of original codes to final themes.**
(TIF)

**S2 Table. Ethics (Local IRB) approval history for the PROSYNK trial.**
(TIF)

**S3 Table. Regulatory and sponsor approval history for the PROSYNK trial.**
(TIF)

## Acknowledgments

The authors gratefully acknowledge the contributions of all PROSYNK trial participants and the dedicated staff at the Kenya Medical Research Institute (KEMRI)/Centre for Global Health Research in Kisumu, Kenya, for their efforts in data collection and trial implementation.

## Author contributions

**Conceptualization:** Micah J. June, Dónal O'Mathúna.

**Data curation:** Micah J. June, Stephen J. Allen.

**Formal analysis:** Micah J. June, Nicodemus O. Mbanda, Polland O. Miruye, Dónal O'Mathúna.

**Funding acquisition:** Micah J. June, Dónal O'Mathúna.

**Investigation:** Micah J. June.

**Methodology:** Micah J. June, Mary I. Otiti, Stephen J. Allen, Dónal O'Mathúna.

**Project administration:** Micah J. June.

**Resources:** Stephen J. Allen.

**Supervision:** Stephen J. Allen, Dónal O'Mathúna.

**Validation:** Micah J. June, Nicodemus O. Mbanda, Polland O. Miruye, Stephen J. Allen, Dónal O'Mathúna.

**Visualization:** Micah J. June.

**Writing – original draft:** Micah J. June.

**Writing – review & editing:** Micah J. June, Mary I. Otiti, Nicodemus O. Mbanda, Polland O. Miruye, Stephen J. Allen, Dónal O'Mathúna.

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
