## [Decision Letter · Decision Letter 0]

15 Aug 2025

PGPH-D-25-01630

The experience of conducting non-pandemic research during the COVID-19 pandemic: a case study

Dear Dr. O’Mathúna,

Thank you for submitting your manuscript to PLOS Global Public Health. After careful consideration, we feel that it has merit but does not fully meet PLOS Global Public Health’s publication criteria as it currently stands. Therefore, we invite you to submit a revised version of the manuscript that addresses the points raised during the review process.

In particular, please ensure you answer carefully Reviewer 2’s (isn’t it always reviewer two?) comment about the thinness of the ethical analysis in the paper. If you feel that another framework is better suited for the paper, that’s fine too. Robustness and depth of analysis in terms of a theoretical framework is what we’re looking for. 

We look forward to receiving your revised manuscript.

Kind regards,

Diego S Silva, PhD

Academic Editor

Journal Requirements:

i. State the initials, alongside each funding source, of each author to receive each grant.

ii. State what role the funders took in the study. If the funders had no role in your study, please state: “The funders had no role in study design, data collection and analysis, decision to publish, or preparation of the manuscript.”

2. Please ensure that your Ethics Statement is available in its entirety at the beginning of your Methods section, under a subheading 'Ethics Statement'.

3. Please provide separate figure files in .tif or .eps format.

4. In the online submission form, you indicated that “Data supporting the findings of this study are available upon reasonable request. Interested parties should contact the corresponding author via email. Access will be granted following approval by all principal investigators involved in the original study.”.

3. Uploaded as supplementary information.

Additional Editor Comments (if provided):

Reviewers' comments:

Reviewer's Responses to Questions

**Comments to the Author**

1. Does this manuscript meet PLOS Global Public Health’s publication criteria?

Reviewer #1: Yes

Reviewer #2: Yes

2. Has the statistical analysis been performed appropriately and rigorously?

Reviewer #1: N/A

Reviewer #2: N/A

3. Have the authors made all data underlying the findings in their manuscript fully available (please refer to the Data Availability Statement at the start of the manuscript PDF file)?

Reviewer #1: Yes

Reviewer #2: No

4. Is the manuscript presented in an intelligible fashion and written in standard English?

Reviewer #1: Yes

Reviewer #2: Yes

Reviewer #1: This is a valuable manuscript that describes the implementation of a non-COVID-19 related trial during the COVID-19 pandemic in Kenya. The manuscript is mainly based on a review of trial-related documents, including meeting minutes, supplemented with review of the literature and other documents with information on wider pandemic response. The authors report on the research regulatory and operational challenges encountered by the trial including how the trial sponsors, investigators and oversight committees adapted the trial to ensure that it was implemented within the restrictions of the COVID-19 pandemic, and in scientifically valid and ethical manner.

It would be helpful for the authors to highlight the challenges encountered by the trial on account of being non-pandemic research conducted during a pandemic. This would be a unique addition to the extensive literature on ethical and practical research challenges during public health emergencies. Also, readers would benefit from additional information about the research team, methodology and context of the study. For example, reference is made of “significant delays” and “expedited approvals” without information on the expected timelines, other than study completion which was delayed by seven months. Specific comments follow.

Title and Abstract

1. Given that the manuscript is manly based on document review, rather than other forms of qualitative research methods such as interviews, consider whether titling this as a “…description of conducting…” might be more appropriate than framing it as “The experience of conducting…” There is limited reporting of direct experience or views related to conducting the trial.

2. Lines 24 and 26-27: What are the “ethics and regulatory submissions” that were reviewed? Specifically, it’s unclear whether authors reviewed the comments/feedback that the trial received from the ethics review committee.

3. Line 27: Consider deleting “literature review” as one of the methods used to generate findings reported in this manuscript. This is because the approach to literature review is not described in the Methodology or outcomes presented in the Results.

4. Consider adding participant observation as one of the methods used to generate findings. It is indicated (Lines 102 -104) that the first author had direct experience of trial implementation and that this experience contributed (Lines 97-98) to findings reported in this manuscript.

Introduction

5. Lines 54-56: The authors may consider expanding this section to clarify the key ethical and practical issues raised by the conduct of non-pandemic research during a pandemic. For instance, what is the ethical justification for conducting non-pandemic research during a pandemic.

6. Throughout the manuscript, there is mention of the need for ethical research, but it’s unclear how ethics or ethical issues are conceptualized. The discussion seem to focus on apparent practical challenges, such as staff shortages, mobility restriction, with little evidence of ethics analysis.

7. Highlight that the trial protocol was submitted and received initial ethics approval before the COVID-19 pandemic (October 2019), and it’s only amendments and submission to regulators that were considered during the pandemic.

Methods

8. Besides the first author, were any of the co-authors involved in the PROSYNK trial in any capacity? A fuller description of the research team, including roles authors in data collection and analysis, would help readers appreciate the insights, skills and experiences underlying the findings presented.

9. It would be great to describe the ethics and regulatory approval process for trials in Kenya during pandemic and non-pandemic periods, including expected turnaround times. This would provide a background for later discussions around regulatory/ethics delay and expedited approvals.

10. Besides Document Access Log, were any software or platforms used for data management or analysis?

Results

11. There was a half-year delay between initial ethics approval and submission to regulatory authority as co-authors needed to amend protocol following change in the investigational product (Lines 149-150). Can the authors provided more information on why submitting this amendment took that long?

12. How long did the trial investigators take to respond to comments/feedback from ethics committees and regulators? Overall timelines are given – from submission to approvals – but the contribution of trial investigators to review time is unclear.

Reviewer #2: This manuscript presents a retrospective documentary case analysis of the challenges faced by researchers when attempting to conduct a randomized controlled clinical trial (PROSYNK) investigating the effects of probiotics and synbiotics on infant gut health and systemic inflammation in a rural setting in western Kenya during the COVID-19 pandemic. The authors examine the ethical, operational, and logistical barriers encountered, as well as the adaptive strategies implemented to safeguard participant welfare and maintain study integrity during this public health emergency.

STRENGTHS

The research question is highly relevant and timely, addressing the underexplored topic of sustaining non-pandemic research during a global health crisis, particularly in low- and middle-income countries (LMICs). The manuscript is generally well written, logically structured, and free of superfluous content.

The methodological approach, a retrospective documentary case analysis, is well suited to the study’s objectives. The authors reviewed all available trial documents, including ethics submissions, meeting minutes, and progress reports, ensuring comprehensive coverage. Data analysis was conducted using dual coding, with a reported Cohen’s Kappa ≥ 0.6, which strengthens the reliability of findings. The use of a thematic framework analysis is appropriate and rigorously applied.

The results are detailed and supported by quantitative trial performance indicators. Interpretation is solidly grounded in the data and enriched by comparisons with existing literature. The manuscript effectively emphasizes ethical responsiveness, community engagement, and participant protection, and it offers concrete, practical recommendations of clear value for sustaining clinical research during public health emergencies in LMICs.

MAJOR ISSUES

Two substantive issues must be addressed before the manuscript is suitable for publication:

a. Data sharing: In line with PLOS ONE’s recommendations, the authors should deposit the raw data in a publicly accessible repository. In this case, “raw data” should include the original taxonomy of themes from the first coding round and the final taxonomy after refinement.

b. Absence of a developed theoretical framework: While the authors indicate an intention to ground their ethical discussion in principlism and public health ethics frameworks, this effort is underdeveloped and unconvincing. Beyond a brief, perfunctory reference in the Discussion (a single sentence, lines 257 - 259), there is no substantive engagement with principlism or any other explicit conceptual or theoretical framework to guide the analysis of adaptive strategies. Such absence limits the manuscript’s theoretical depth, resulting in a largely descriptive rather than analytically structured account. The authors should consider adopting a well-defined framework, such as resilience theory, implementation science, or a health systems framework, to organize complex ideas, connect findings to prior scholarship, and interpret results in context. A committed application of such a framework would require clearly justifying its selection, summarizing its core principles, and applying it consistently and rigorously to interpret findings, showing how results support, challenge, or expand upon the chosen theoretical perspective. This would situate the study within the broader academic conversation, establish a coherent theoretical basis, and yield more meaningful, generalizable conclusions.

SUGGESTIONS FOR IMPROVEMENT

Abstract

• Explicitly state that this is a case study of the PROSYNK Trial, which assessed the impact of probiotics and synbiotics on infant gut health and systemic inflammation in western Kenya.

Study design

• Strengthen the justification for using a single case study and discuss how findings may be transferable to other settings.

Sampling

• Clarify how documents were included or excluded.

Data collection

• Describe how triangulation was achieved across document types.

Data quality and trustworthiness

• Briefly explain how potential biases arising from the authors’ direct involvement in the trial (lines 102 – 103) were mitigated.

Data analysis

• Provide more detail on the framework guiding coding and indicate any software used (NVIVO, ATLAS.ti).

Results

• Discuss how specific adaptive measures, particularly the shift to remote follow-up in the control arm, could have affected trial validity.

Tables

• Move Supplementary Table 2 (framework analysis of key themes) into the main text. Streamline it by merging the third column (“Key Questions and Responses”) with the second, as the separation is unnecessary.

Figures

• Consider simplifying figures and aligning them more closely with key findings.

• Evaluate whether Supplementary Figure 2 (Conceptual Framework) is necessary. In its current form it is overly complex, unreferenced in the main text, and potentially dispensable.

Discussion

• Consider perhaps adding the following recent reference at the end of line 56, line 233, and line 277: Salamanca-Buentello F, Katz R, Silva DS, Upshur REG, Smith MJ (2024). Research ethics review during the COVID-19 pandemic: An international study. PLoS ONE, 19(4): e0292512. https://doi.org/10.1371/journal.pone.0292512

Ethical, social, and cultural considerations

• Clarify how community input influenced adaptive strategies and how equity considerations were addressed in shifting to remote follow-up for the control arm.

Limitations

• Add a dedicated section acknowledging study limitations.

Contributions to knowledge

• Explicitly link recommendations to global guidance, particularly WHO and CIOMS guidelines on pandemic preparedness and research ethics during public health emergencies. The authors cite a key reference, the 2020 WHO report, but only in the Introduction.

Conclusion

• Address more explicitly the generalizability of findings beyond the PROSYNK context.

MINOR ISSUES

• Line 179: “Peer-reviewed studies and media reports …” Please provide the references for the sources mentioned.

• Line 182: “…a news article dated 8 June 2021…” Again, please provide the appropriate reference.

• Line 246: Change “underscored” to “underscoring”, as the gerund is the correct form here.

• Line 299: “…Recruitment to target, high participant retention rates…” Add AND between “target” and “high” and remove the comma.

• Note that, in the PDF file received, Fig. 1 is out of place, as it appears after Figs. 2 and 3.

• If Supplementary Figure 2 is retained, correct “Participants safety and recruitment” to “Participants’ safety and recruitment”, as the apostrophe is needed to form the possessive.

SUMMARY

This is a valuable and well-executed case study that contributes to the literature on conducting non-pandemic research during public health emergencies in LMICs. Addressing the major concerns, particularly data sharing and strengthening the theoretical framing, will significantly enhance the paper’s analytical depth and policy relevance. With these revisions, the manuscript could offer important lessons for researchers, ethics review committees, and policymakers worldwide.

**Do you want your identity to be public for this peer review?** For information about this choice, including consent withdrawal, please see our Privacy Policy

Reviewer #1: No

Reviewer #2: No

---

## [Decision Letter · Decision Letter 1]

9 Dec 2025

Conducting non-pandemic research during the COVID-19 pandemic: a case study and documentary review

PGPH-D-25-01630R1

Dear Dr. O’Mathúna,

We are pleased to inform you that your manuscript 'Conducting non-pandemic research during the COVID-19 pandemic: a case study and documentary review' has been provisionally accepted for publication in PLOS Global Public Health.

Best regards,

Julia Robinson

Executive Editor

Reviewer Comments (if any, and for reference):

Reviewer's Responses to Questions

**Comments to the Author**

Reviewer #2: All comments have been addressed

publication criteria?

Reviewer #2: Yes

3. Has the statistical analysis been performed appropriately and rigorously?

Reviewer #2: N/A

4. Have the authors made all data underlying the findings in their manuscript fully available (please refer to the Data Availability Statement at the start of the manuscript PDF file)?

Reviewer #2: Yes

5. Is the manuscript presented in an intelligible fashion and written in standard English?

Reviewer #2: Yes

Reviewer #2: The authors have responded thoroughly and constructively to all my previous comments, and I appreciate the considerable time and care they invested in strengthening the manuscript. The revised version demonstrates clear improvements in structure, clarity, and methodological detail, and the expanded discussion of the ethical and operational dimensions of conducting non-pandemic research during COVID-19 is substantially more robust. The supplementary materials now provided further enhance transparency and support the credibility of the analysis.

While I remain only partially satisfied with the depth of integration between the study’s findings and the theoretical frameworks invoked, particularly the application of resilience theory, the authors have nevertheless made meaningful progress. The initial single-sentence treatment has been replaced with coherent paragraphs that situate the findings within resilience theory, articulate the relevance of principlism with greater nuance, and establish clearer links to the ethical and operational challenges encountered. Although the integration remains more descriptive than analytically deep, it no longer appears perfunctory. Given the applied nature of the manuscript and its primary focus on documenting operational experience, the current theoretical engagement is adequate and does not detract from the paper’s contribution. On balance, the theoretical foundations are sufficiently solid for publication, and the manuscript meets the journal’s standards for methodological and ethical rigour.

**Do you want your identity to be public for this peer review?** For information about this choice, including consent withdrawal, please see our Privacy Policy

Reviewer #2: No
